# Antioxidant and Antimelanogenic Activities of *Lactobacillus kunkeei* NCHBL-003 Isolated from Honeybees

**DOI:** 10.3390/microorganisms12010188

**Published:** 2024-01-17

**Authors:** Yeon-Ji Lee, Joo-Hee Choi, Kyung-Ku Kang, Soo-Eun Sung, Sijoon Lee, Minkyoung Sung, Min-Soo Seo, Jong-Hwan Park

**Affiliations:** 1Preclinical Research Center, Daegu-Gyeongbuk Medical Innovation Foundation, Daegu 41061, Republic of Korea; wudsl0000@kmedihub.re.kr (Y.-J.L.); cjh522@kmedihub.re.kr (J.-H.C.); kangkk@kmedihub.re.kr (K.-K.K.); sesung@kmedihub.re.kr (S.-E.S.); sjlee1013@kmedihub.re.kr (S.L.); tjdalsrud27@kmedihub.re.kr (M.S.); 2Laboratory of Veterinary Tissue Engineering, College of Veterinary Medicine, Kyungpook National University, 80 Daehak-ro, Buk-gu, Daegu 41566, Republic of Korea; 3Laboratory Animal Medicine, College of Veterinary Medicine and the Brain Korea 21 PLUS Project Team, Chonnam National University, 77 Yongbong-ro, Buk-gu, Gwangju 61186, Republic of Korea

**Keywords:** *Lactobacillus kunkeei* NCHBL-003, honeybee microbiome, postbiotic supernatants, antioxidant, antimelanogenic

## Abstract

Excessive reactive oxygen species production can detrimentally impact skin cell physiology, resulting in cell growth arrest, melanogenesis, and aging. Recent clinical studies have found that lactic acid bacteria have a special effect directly or indirectly on skin organs, but the exact mechanism has not been elucidated. In this study, we investigated the mechanisms underlying the antioxidant protective effect and the inhibitory effect on melanin synthesis of *Lactobacillus kunkeei* culture supernatant (CSK), isolated from *Apis mellifera* Linnaeus (the Western honeybee). CSK exhibited notable efficacy in promoting cell migration and wound healing under oxidative stress, surpassing the performance of other strains. CSK pretreatment significantly upregulated the expression of Nrf2/HO-1 (nuclear factor erythroid 2-related factor 2/heme oxygenase-1), a key player in cellular defenses against oxidative stress, relative to the control H_2_O_2_-treated cells. The DCF-DA (dichloro-dihydro-fluorescein diacetate) assay results confirmed that CSK’s ability to enhance Nrf2 and HO-1 expression aligns with its robust ability to remove H_2_O_2_-induced reactive oxygen species. Furthermore, CSK upregulated MAPK (mitogen-activated protein kinase) phosphorylation, an upstream signal for HO-1 expression, and MAPK inhibitors compromised the wound-healing effect of CSK. Additionally, CSK exhibited inhibitory effects on melanin synthesis, downregulating melanogenesis-related genes in B16F10 cells. Thus, the present study demonstrated that CSK exhibited antioxidant effects by activating the Nrf2/HO-1 pathway through MAPK phosphorylation, thereby restoring cell migration and demonstrating inhibitory effects on melanin production. These findings emphasize the antioxidant and antimelanogenic potential of CSK, suggesting its potential use as a therapeutic agent, promoting wound healing, and as an active ingredient in skin-lightening cosmetics.

## 1. Introduction

The skin functions as a primary barrier, protecting the human body against microbes, ultraviolet radiation, and environmental pollutants. Exposure to these stimuli causes damage to keratinocytes, the outermost skin cells, triggering the production of reactive oxygen species (ROS) [1,2]. Low-level ROS production is pivotal in numerous physiological activities, serving as an essential component of intercellular signaling pathways, including oxygen metabolism, inflammation, and immune responses [3,4]. However, heightened oxidative stress due to excessive ROS production halts the cell cycle. This interruption prevents the migration or proliferation of cells and eventually leads to mechanisms for programmed cell death (i.e., apoptosis) [5,6,7,8,9]. In melanocytes, oxidative stress promotes melanin biosynthesis, which is regulated by tyrosinase (TYR), microphthalmia-associated transcription factor (MITF), tyrosinase-related protein 1 (TYRP1), and TYRP2 [10,11]. These intricate processes collectively contribute to skin aging.

In response to oxidative stress, nuclear factor erythroid 2-related factor 2 (Nrf2), which is constantly degraded by Keap1 (Kelch-like ECH-associated protein 1), accumulates in the nuclei and triggers the production of antioxidant proteins, including GCL (glutamate-cysteine ligase), GSTs (glutathione S-transferases), NQO1 (NAD(P)H:quinone reductase-1), and heme oxygenase-1 (HO-1) [12,13,14]. This production is induced by the binding of Nrf2 to the ARE (antioxidant response element)—an enhancer sequence located in the promoter [15]. Among these antioxidative proteins, HO-1 plays a significant role in mitigating oxidative stress by catabolizing free hemes. Heme-catalyzed products exhibit cytoprotective, anti-inflammatory, antioxidant, and bactericidal properties [16,17]. HO-1 also promotes wound healing by stimulating the proliferation of keratinocytes [18]. Given these properties, the Nrf2/HO-1 pathway is considered a promising new therapeutic target for mitigating oxidative stress-related diseases [19,20].

Mitogen-activated protein kinases (MAPKs) are proteins that mediate signal transduction in response to external stimuli, participating in cellular responses, such as cell proliferation, growth, movement, and survival [21]. In higher organisms, MAPKs encompass extracellular signal-regulated protein kinase (ERK), c-jun N-terminal kinase (JNK), and p38 kinase, which are phosphorylated in response to growth factors, oxidative stress, and inflammatory cytokines [22]. These phosphorylated MAPKs then phosphorylate target transcription factors. Several studies have demonstrated the involvement of MAPKs in Nrf2/HO-1 signaling [23]. Activated MAPKs induce the translocation of Nrf2 into the nucleus, enhancing oxidative stress mitigation [24,25].

Lactic acid bacteria (LAB), Gram-positive bacteria that convert sugar to lactic acid, generate numerous beneficial components [26,27] with anti-inflammatory and antibacterial properties, suppressing the proliferation of harmful bacteria [28]. Researchers are increasingly interested in the treatment of skin wounds using LAB as a treatment, with fewer side effects on the skin. Recent studies have shown that *Lactobacillus rhamnosus* GG and *Lactobacillus gasseri*, which have been proven to increase gastric barrier repair, accelerate wound healing in primary human keratinocytes [29]. *Lactobacillus plantarum* supernatant (CSP) has been shown to provide 60–100% protection against H_2_O_2_-induced stress in A549 cells; it has also been shown to promote wound healing more quickly than a positive control group in mice [30]. For centuries, bee products, such as honey, propolis, royal jelly, and bee pollen, have been employed to enhance human health and address skin issues [31,32,33,34,35] owing to their anti-inflammatory, anticancer, and antioxidant properties. These effects are related to the microbiota in bee habitats and digestive systems. In particular, several studies have highlighted the benefits of honeybee-associated LAB [36,37,38,39,40]. While most of these studies have demonstrated the effectiveness of probiotics in protecting hosts [41,42], their ability to safeguard the skin or inhibit melanin synthesis remains unexplored. This study explored the antioxidative and antimelanogenic properties of *Lactobacillus kunkeei* culture product on skin cells, shedding light on the skin protection potential of honeybee-associated LAB.

## 2. Materials and Methods

### 2.1. Reagents

Fetal bovine serum (FBS), Dulbecco’s Modified Eagle’s Medium (DMEM), phosphate-buffered saline, and other culture reagents were obtained from Gibco (Grand Island, NY, USA). Anti-phospho-JNK, anti-phospho-p38, anti-phospho-ERK, anti-Nrf2, and anti-HO-1 antibodies were obtained from Cell Signaling Technology (Danvers, MA, USA). Anti-Nrf2, anti-α-tubulin, anti-laminB1, and β-actin antibodies were purchased from Santa Cruz Biotechnology (Dallas, TX, USA). SP600125 (JNK inhibitor), SB203580 (p38 inhibitor), and PD98059 (ERK inhibitor) were purchased from Selleck Chemicals (Houston, TX, USA). The EveryBlot Blocking Buffer was purchased from BioRad (Hercules, CA, USA), and the enhanced chemiluminescence substrate was procured from Dogenbio (Seoul, Republic of Korea). TRIzol reagent and Halt Protease and Phosphatase Inhibitor Cocktail (100×) were purchased from Thermo Fisher (Waltham, MA, USA), and all other chemicals were purchased from Sigma-Aldrich (St. Louis, MO, USA).

### 2.2. Isolation and Identification of Gut Bacteria in Honeybees

The honeybees (*Apis mellifera* Linnaeus) were collected from three sites in Gwangju Metropolitan City, Republic of Korea. The sampling sites were located on the south side of the city. Twenty-five workers (free-flying bees) were collected from each colony at three different sites. All collections were carried out in June 2022, and samples were stored on ice before bacterial isolation. Fresh honeybee samples were surface-sterilized three times with 7% sodium hypochlorite in sterile plates. The bee midguts were removed with a sterile blade and homogenized in 1 mL of phosphate-buffered saline. The gut homogenate (100 μL) was spread, using the standard dilution method, onto MRS (de Man, Rogosa, and Sharpe) agar medium (Oxoid, Hampshire, UK). The isolates were incubated for 3–4 days, at 37 °C, under anaerobic conditions, using anaerobic jars (BD, Franklin Lakes, NJ, USA). For the acquisition of pure bacterial isolates, 100 colonies with different morphologies were collected and subcultured. Gram staining and biochemical characterizations (catalase, citrate, H_2_S, indole test, methyl red test, oxidase, urease, and Voges–Proskauer test) were conducted as initial screening for *Lactobacillus*. Gram-positive and catalase-negative bacilli were selected. The isolates were maintained at −80 °C. A DNA extraction kit (QIAGEN, Hilden, Germany) was used to perform a modified DNA extraction protocol. Genomic DNA was prepared from all thirty bacterial isolates. The 16S rDNA gene (1500 bp) was amplified using 27F and 1492R primers (5′-AGAGTTTGATCCTGGCTCAG-3′ and 5′-GGTTACCTTGTTACGACTT-3′, respectively), targeting lactobacilli at the genus level. The polymerase chain reaction (PCR) amplifications consisted of initial denaturation at 95 °C for 60 s, followed by 35 cycles of 95 °C for 60 s, 55 °C for 60 s, and 72 °C for 60 s. The PCR fragment was sequenced according to the manufacturer’s instructions for the BigDye Terminator Cycle Sequencing Kit, using a DNA sequencer (ABI 3730; Applied Biosystems, Foster City, CA, USA). DNA sequences obtained from the study were submitted to the DDBJ (DNA Data Bank of Japan) database. Species identification was conducted through a homology search of the 16S rRNA gene sequences, using the BLAST database on the NCBI website. Subsequently, the strains were designated as *Lactobacillus kunkeei* NCHBL-003, *Lactobacillus plantarum* NCHBL-004, and *Lactobacillus reuteri* NCHBL-005.

### 2.3. Preparation of the Culture Supernatants of Lactobacillus *spp.*

Probiotic strains *L. kunkeei* NCHBL-003, *L. plantarum* NCHBL-004, and *L. reuteri* NCHBL-005 were individually isolated from the digestive tracts of honeybees. In our experiments, each strain was used independently, without any combination treatments. For bacterial culture preparations, single colonies were inoculated into 10 mL of MRS broth and grown overnight at 37 °C, with shaking. A 1:10 dilution of the bacterial suspension was cultured, with shaking, until it reached an optical density of 0.6 at 600 nm, corresponding to 1 × 10^9^ colony-forming units (CFU) mL^−1^. In the preparation of cell-free culture supernatants for the in vitro assays, these *Lactobacillus* strains were grown in DMEM at 37 °C, with shaking, for 24 h. The culture supernatants were collected via centrifugation at 10,000× *g* for 10 min, adjusted to pH 7.4, and then sterilized using a syringe filter (0.22 μm). DMEM without bacterial culture served as a negative control.

### 2.4. Cell Culture

The human keratinocyte cell line HaCaT (Cytion, Eppelheim, Germany) and the mouse melanoma cell line B16F10 (ATCC, Rockville, MD, USA) were cultured in DMEM supplemented with 10% (*v*/*v*) FBS and 1% (*v*/*v*) penicillin–streptomycin (P/S), and they were maintained at 37 °C in a 5% CO_2_-humidified incubator.

### 2.5. Cell Viability Assay

HaCaT cells (1 × 10^5^ cells/mL/well) were seeded into a 96-well plate containing DMEM supplemented with 10% (*v*/*v*) FBS and incubated at 37 °C for 16 h. Subsequently, the cells were treated with 2.5%, 5%, or 10% (*v*/*v*) *Lactobacillus* spp. culture supernatant for 24 h. Then, the cells were treated with MTT (3-(4,5-dimethylthiazol-2-yl)-2,5-diphenyltetrazolium bromide) reagent (4 mg/mL in DMEM) for 4 h. The resulting formazan crystals were dissolved in DMSO (dimethyl sulfoxide), and their absorbance was measured at 540 nm, using a microplate reader.

### 2.6. Wound Healing Assay

HaCaT cells (3 × 10^5^ cells/mL/well) were seeded into a 12-well plate (with 2-well silicone inserts) containing DMEM supplemented with 10% (*v*/*v*) FBS and 1% (*v*/*v*) P/S and incubated at 37 °C for 16 h. Subsequently, the medium was replaced with FBS-free DMEM, and the silicone inserts were removed to create a cell-free gap. Then, the medium was replaced with DMEM containing 1% (*v*/*v*) FBS and supplemented with 2.5%, 5%, or 10% (*v*/*v*) *Lactobacillus* spp. culture supernatant. After 1 h, the cells were treated with 500 μM of H_2_O_2_, and images of the cells were captured after 24 h. The area of the cell-free gap was quantified using ImageJ software.

### 2.7. ROS Generation

HaCaT cells (1 × 10^5^ cells/mL/well) were seeded into a 12-well plate containing DMEM supplemented with 10% (*v*/*v*) FBS and 1% (*v*/*v*) P/S and incubated at 37 °C for 16 h. Subsequently, the medium was replaced with DMEM containing 1% (*v*/*v*) FBS and supplemented with 2.5%, 5%, or 10% (*v*/*v*) *L. kunkeei* NCHBL-003 culture supernatant (CSK). After 1 h, the cells were treated with 500 μM H_2_O_2_ and incubated at 37 °C for 3 h. Subsequently, the cells were stained with 10 μM dichloro-dihydro-fluorescein diacetate (DCF-DA) for 20 min, and the generated ROS were visualized via fluorescence microscopy. The images were quantified using ImageJ (National Institutes of Health, Bethesda, MD, USA).

### 2.8. Melanin Generation

B16F10 cells (1 × 10^5^ cells/mL/well) were seeded into a 12-well plate containing DMEM supplemented with 10% (*v*/*v*) FBS and 1% (*v*/*v*) P/S and cultured at 37 °C for 24 h to allow for attachment and stabilization. Subsequently, the culture medium was replaced with phenol red-free culture medium (Hyclone, Cytiva, Marlborough, MA, USA). Then, the cells were treated with 2.5%, 5%, or 10% CSK or with 2% arbutin. After 1 h, the cells were stimulated with α-melanocyte-stimulating hormone (α-MSH). After 72 h, the melanin levels in the medium were measured at 472 nm, using a microplate reader.

### 2.9. Protein Extraction and Western Blotting

HaCaT cells (1 × 10^5^ cells/mL/well) were seeded onto a 12-well plate containing DMEM supplemented with 10% (*v*/*v*) FBS and 1% (*v*/*v*) P/S and incubated at 37 °C for 16 h. After the treatment for each experiment, the cells were lysed at the indicated time points in a buffer containing 1% Nonidet P-40, 50 mM Tris (pH 7.4), 250 mM NaCl, 5 mM EDTA, 50 mM NaF, 1 mM Na_3_VO_4_, and 0.02% NaN_3_, supplemented with Halt Protease and Phosphatase Inhibitor Cocktail (100×) and 2 mM dithiothreitol. In a parallel experiment, nuclear and cytoplasmic proteins were extracted using the NE-PER Nuclear and Cytoplasmic Extraction Kit (Pierce, Rockford, IL, USA), following the manufacturer’s instructions. Protein samples were separated via 10% SDS–PAGE (sodium dodecyl sulfate–polyacrylamide gel electrophoresis) and transferred onto nitrocellulose membranes. The membranes were blocked with EveryBlot Blocking Buffer for 1 h at room temperature and probed with primary antibodies against phospho-JNK, phospho-p38, phospho-ERK, Nrf2, HO-1, and β-actin overnight at 4 °C. Subsequently, the membranes were incubated with relevant secondary antibodies for 2 h at room temperature, and the proteins were detected using the enhanced chemiluminescence substrate.

### 2.10. Immunofluorescence Analysis

HaCaT cells (3 × 10^5^ cells/mL/well) were seeded into an eight-well chamber containing DMEM medium supplemented with 10% (*v*/*v*) FBS and 1% (*v*/*v*) P/S and incubated at 37 °C for 16 h. After treatment with CSK or H_2_O_2_, the cells were fixed using a 4% formaldehyde solution and permeabilized using 0.2% TritonX-100 (Sigma, St. Louis, MO, USA). Subsequently, the cells were blocked with 3% bovine serum albumin and then incubated with an anti-Nrf2 antibody (1:50 in 3% bovine serum albumin) for 2 h at room temperature. Then, fluorescein isothiocyanate-conjugated anti-mouse secondary antibody (Sigma, St. Louis, MO, USA) was added, and the cells were incubated for 2 h at room temperature. The cells were mounted onto the cover glass, using ProLong Gold Antifade Mountant with DNA Stain DAPI (Thermo Fisher, Waltham, MA, USA), and observed and photographed using a confocal microscope (Leica, Wetzlar, Germany). The exposure time was consistent across samples in all experiments.

### 2.11. Quantitative Real-Time PCR

Total RNA from HaCaT and B16F10 cells was extracted using the TRIzol method. Briefly, cells were lysed using TRIzol reagent and then treated with chloroform to induce phase separation. The aqueous phase containing RNA was collected and precipitated using isopropanol. The RNA pellet was washed with ethanol, air-dried, and then dissolved in RNase-free water. The quality and quantity of the extracted RNA were assessed using a NanoDrop. The RNA was reverse transcribed to cDNA, using the ReverTra Ace qPCR RT Master Mix (TOYOBO BioTechnology, Osaka, Japan), following the manufacturer’s instructions. Quantitative real-time (qPCR) was performed using QGreen 2X SYBR Green PCR Master Mix (CellSafe, Yongin, Republic of Korea) and gene-specific primers (Appendix A).

### 2.12. Statistical Analysis

Data were statistically analyzed using Student’s *t*-test. Calculations to determine statistically significant differences between the experimental and control groups were performed using GraphPad Prism, version 5.01 (GraphPad Software, San Diego, CA, USA). Statistical significance was acknowledged when *p* < 0.05.

## 3. Results

### 3.1. CSK Promotes Cell Migration during Oxidative Stress in HaCaT Cells

We isolated *L. kunkeei* NCHBL-003, *L. plantarum* NCHBL-004, and *L. reuteri* NCHBL-005 from honeybee gut microbiota and investigated the potential of their culture supernatants (CSK, CSP, and CSR, respectively) in promoting cell migration during oxidative stress through an in vitro wound healing assay. HaCaT cells with a cell-free gap were pretreated with various concentrations (2.5%, 5%, or 10%) of each culture supernatant for 1 h, and then oxidative stress was induced using H_2_O_2_. While H_2_O_2_-induced oxidative stress hindered cell migration and proliferation, pretreatment with *Lactobacillus* spp. culture supernatants enhanced wound healing in the oxidative environment (Figure 1A,B). Notably, CSK was associated with the highest recovery of migration ability under oxidative stress. Moreover, cytotoxicity assessments using MTT assays revealed that the *Lactobacillus* spp. culture supernatants did not have adverse effects on HaCaT cells (Figure 1C). Therefore, subsequent experiments were focused on elucidating the antioxidant mechanism of CSK, which demonstrated the most effective recovery from oxidative stress-induced impaired cell migration among the studied *Lactobacillus* spp.

### 3.2. CSK Inhibits ROS Generation by Modulating the Nrf2/HO-1 Pathway

Given that CSK mitigated the deleterious effects of H_2_O_2_ on wound healing, we investigated whether CSK is involved in inhibiting H_2_O_2_-induced ROS production. To determine the protective effects of CSK against H_2_O_2_-induced oxidative stress, we measured intracellular ROS generation in HaCaT cells, using the DCF-DA assay. The green fluorescence signals of DCF were stronger in H_2_O_2_-treated cells than in untreated cells, indicating that H_2_O_2_ induced abundant ROS production in HaCaT cells. However, pretreatment with 2.5%, 5%, and 10% CSK was associated with significant attenuation of the increase in DCF fluorescence intensity (Figure 2A,B), demonstrating CSK’s antioxidant capacity. Generally, Nrf2/HO-1 activity increases under H_2_O_2_-induced oxidative stress due to its role as an antioxidant [43]. To confirm that CSK protected cells from oxidative stress through Nrf2/HO-1 activity, we induced oxidative stress in CSK-treated HaCaT cells, using H_2_O_2_, and assessed the expression of Nrf2 and HO-1, using Western blot analyses and qPCR. H_2_O_2_ treatment increased the mRNA and protein levels of Nrf2 and HO-1 in HaCaT cells, indicating that oxidative stress activated the Nrf2/HO-1 pathway (Figure 2C,D). The expression of Nrf2 and HO-1 was significantly higher in the presence of CSK than in the sole presence of H_2_O_2_. Furthermore, CSK treatment augmented Nrf2 expression within the nucleus (Figure 2C,E). Thus, CSK amplifies the expression of Nrf2 and HO-1 during oxidative stress, leading to the activation and translocation of Nrf2 from the cytoplasm to the nucleus. These results indicate that CSK exerts its antioxidative effects, at least in part, by activating the Nrf2/HO-1 signaling pathway, amplifying the cellular defense mechanisms against oxidative damage.

### 3.3. CSK Induces Nrf2/HO-1 through MAPK Phosphorylation in HaCaT Cells

Previous studies have reported that MAPKs mediate the activity of antioxidant-related factors, such as Nrf2 and HO-1, in various cell types [24,25]. Therefore, we investigated the potential of CSK to activate MAPKs. HaCaT cells were subjected to treatment with CSK for specified time points (0, 0.5, 1, 3, and 6 h) to assess the dynamic changes in MAPK activation. CSK treatment induced the strong phosphorylation of JNK, p38, and ERK at 0.5 h after treatment (Figure 3A). The protein expression of Nrf2 and HO-1 increased strongly after 3 h (Figure 3A). Furthermore, the *Nrf2* and *HO*-1 genes showed a strong increase after 1 h and 6 h of CSK treatment, respectively (Figure 3B). Considering that CSK induced the activity of MAPKs at an earlier time point than Nrf2/HO-1, CSK could be expected to regulate Nrf2/HO-1 by activating MAPKs in HaCaT cells. To further verify whether CSK-induced Nrf2/HO-1 expression is related to MAPKs, HaCaT cells were treated with SP600125 (JNK inhibitor), SB203580 (p38 inhibitor), and PD98059 (ERK inhibitor), followed by CSK treatment. The CSK-induced expression of Nrf2 and HO-1 was inhibited by MAPK inhibitors (Figure 3C). Moreover, the addition of inhibitors significantly reduced the expression of Nrf2 and HO-1 mRNA relative to that observed in groups without inhibitors (CSK alone) (Figure 3D). These results indicate that CSK upregulates Nrf2 and HO-1 expression by activating MAPKs.

### 3.4. CSK Promotes Cell Migration through MAPKs during Oxidative Stress in HaCaT Cells

CSK increased the expression of the antioxidant factors Nrf2 and HO-1 through MAPKs; therefore, we further investigated whether CSK promoted cell migration through MAPKs during oxidative stress. HaCaT cells with a cell-free gap were pretreated with SP600125 (JNK inhibitor), SB203580 (p38 inhibitor), and PD98059 (ERK inhibitor) and then treated with CSK. Subsequently, the cells were treated with H_2_O_2_ to induce oxidative stress for 24 h, and the cell-free gap was observed under a microscope. CSK restored cell migration impaired by oxidative stress, and MAPK inhibitors reversed the effect of CSK during oxidative stress (Figure 4A,B). These results indicate that MAPKs mediate the protective effect of CSK against H_2_O_2_-induced impairments in cell migration in HaCaT cells.

### 3.5. CSK Inhibits Melanin Synthesis by Modulating the Expression of Melanogenic Genes in B16F10 Cells

In the skin, oxidative stress leads to cellular damage and promotes melanin biosynthesis [44]. The inhibitory effects of CSK on melanin synthesis were investigated via the treatment of B16F10 cells with CSK at concentrations of 2.5%, 5%, and 10%. First, no CSK-induced cytotoxicity was observed (Figure 5A). When B16F10 cells were stimulated with α-MSH to enhance melanin synthesis, CSK treatment resulted in a concentration-dependent reduction in melanin production (Figure 5B). In particular, the 10% CSK treatment exhibited stronger inhibition of melanin synthesis relative to the positive control group treated with arbutin. Additionally, CSK treatment decreased the mRNA expression levels of TYR, a protein involved in melanin precursor production, and melanogenesis-related genes (*Mitf*, *Tyrp1*, and *Tyrp2*) (Figure 5C). The suppression of mRNA expression by CSK was significant and comparable to the effect of arbutin treatment. These results suggest that CSK not only reduces oxidative stress in the skin but also prevents melanin production by suppressing the expression of melanogenesis-related genes.

## 4. Discussion

The numerous health benefits of probiotics are well recognized, with recent studies particularly emphasizing their positive effects on skin health, such as antioxidant, antipigmentation, and anti-wrinkle properties [45,46,47]. However, the specific mechanisms underlying these beneficial effects on the skin remain poorly understood. The primary objective of our study was to explore the antioxidant and antimelanogenic properties of *L. kunkeei* cultures isolated from the honeybee gut. Our focus was on elucidating the mechanism of action of CSK, particularly its impact on the Nrf2/HO-1 activation pathway through MAPK, aiming to understand its role in wound healing via the inhibition of oxidative stress. Additionally, we examined CSK’s effect on α-MSH-induced melanin synthesis inhibition in melanocytes.

Honeybee products have a long history of use in traditional remedies, and there is increasing scientific interest in the LAB they contain. LAB derived from honeybees exhibit diversity due to environmental factors and individual bee characteristics, with *Lactobacillus* spp. being a predominant genus in the honeybee gut microbiota [48,49]. Consistent with previous reports, we isolated and identified three honeybee-derived LAB, namely *L. kunkeei* NCHBL-003, *L. plantarum* NCHBL-004, and *L. reuteri* NCHBL-005. In this study, *Lactobacillus* culture supernatants (CSK, CSP, and CSR) were utilized, representing a form of postbiotics derived from probiotic fermentation [50]. Postbiotics have gained considerable interest due to their demonstrated efficacy, which is comparable to that of probiotic bacteria, making them increasingly relevant in academic research [51]. Recent studies have explored how to enhance skin repair by using not only live probiotics but also postbiotics, including heat-killed bacteria and their metabolites. It was found that the fermented supernatant of *L. kefiri* can postpone cellular senescence by reducing oxidative stress and the expression of aging-related genes in human skin cells affected by H_2_O_2_ [52]. Furthermore, the culture supernatant of *L. gasseri* BNR17 has been shown to inhibit oxidative stress and hyperpigmentation in skin cells [53]. Oxidative stress, induced by free radicals, is intricately linked to age-related phenomena, including skin wrinkles. Treatment with H_2_O_2_ in cellular models leads to an excess of ROS, disrupting the oxidative balance [54], which is useful for establishing an oxidative stress model to investigate cellular responses in an environment that simulates oxidative stress [55]. Such models are crucial for studying aging-related changes, particularly in skin aging and wrinkle formation.

In this study, we confirmed that pretreatment with CSR and CSK inhibited H_2_O_2_-induced cell migration impairment without causing cytotoxicity. Specifically, CSK reduced the wound area by more than 90%, even at low concentrations. Furthermore, DCF staining confirmed CSK’s effective inhibition of ROS production, which facilitated the recovery of cell migration disrupted by H_2_O_2_. We then proceeded to examine the potential pathways through which CSK exerts its antioxidant action in HaCaT cells.

Numerous studies have established the role of Nrf2/HO-1 signaling in HaCaT cells for cellular protection against oxidative stress, suggesting its relevance in treating various diseases [56,57,58]. To investigate the cytoprotective mechanism of CSK against oxidative stress, we examined its association with Nrf2/HO-1 activation. Remarkably, the H_2_O_2_-induced protein and gene expression of Nrf2 and HO-1 increased significantly following CSK pretreatment. Interestingly, HO-1 expression was more pronounced in the presence of H_2_O_2_ than in the presence of CSK alone. This suggests that CSK further enhances HO-1 activity facilitated by Nrf2 under oxidative stress conditions. In various cell types, the expression of antioxidant enzymes, including Nrf2 and HO-1, is regulated by MAPKs [20,59,60]. In the present study, CSK pretreatment rapidly increased JNK, p38, and ERK phosphorylation and was followed by an upregulation of Nrf2 and HO-1. This consistent pattern suggests that CSK activates MAPK pathways, contributing to enhanced Nrf2 and HO-1 expression in response to oxidative stress. Moreover, MAPK inhibitors reversed the CSK-induced upregulated phosphorylation of JNK, p38, and ERK, attenuating Nrf2/HO-1 expression. Additionally, our findings demonstrated that CSK treatment facilitated the recovery of cell migration impaired by oxidative stress. However, this beneficial effect was reversed when MAPK inhibitors were introduced. These results underscore the crucial role of MAPK signaling in the cytoprotective actions of CSK, emphasizing its potential as a therapeutic target to mitigate oxidative stress-induced cell damage.

Our study demonstrated that CSK effectively reduced α-MSH-induced melanogenesis in melanocytes without causing cytotoxicity, exhibiting efficacy comparable to arbutin—a recognized skin-lightening agent. Ultraviolet-stimulated ROS in skin cells stimulate α-MSH through MC1R, inducing melanin synthesis [61]. Our results showed the increased melanin content of melanocytes following α-MSH treatment. Melanin synthesis in melanocytes primarily involves three key enzymes: TYR, TYRP1, and TYRP2. TYR initiates melanogenesis by converting tyrosine to dopaquinone, while TYRP1 contributes to the later stages, including DHICA (5,6-dihydroxyindole-2-carboxylic acid) formation. TYRP2 also influences melanogenesis. Thus, all three of these enzymes play central roles in regulating skin pigmentation [62]. Additionally, MITF, a nuclear transcription factor, regulates both gene expression and pigment production in melanocytes [63], akin to TYR, TYRP1, and TYRP2. Melanin overproduction in melanocytes contributes to various skin conditions, necessitating the exploration of effective agents to regulate pigmentation [64]. Arbutin, a cosmetic ingredient, lightens skin by inhibiting TYR via its conversion to hydroquinone, a compound known for its depigmenting effects [65]. Consistent with previous studies, our findings show that arbutin treatment reduced melanin secretion and the mRNA expression of *tyr*, *mitf*, *tyrp1*, and *tyrp2* in the presence of α-MSH, a major stimulator of melanin biosynthesis in melanocytes. Furthermore, CSK treatment yielded a similar effect to arbutin, reducing melanin secretion and the mRNA expression of *tyr*, *mitf*, *tyrp1*, and *tyrp2* in a concentration-dependent manner.

While our study successfully validated the antioxidant and antimelanogenic effects of the investigated probiotic culture media, we did not conduct a comprehensive analysis of the individual components directly responsible for these observed effects. The primary focus of our investigation was assessing the overall efficacy of probiotic culture media rather than identifying specific bioactive compounds. Therefore, while our results provide robust evidence for the beneficial effects of the investigated culture media, further studies are warranted to elucidate the precise molecular constituents contributing to the observed antioxidant and antimelanogenic properties. To address these limitations in our future experiments, we will focus on unveiling the precise mechanisms underlying the CSK effect. This will involve employing techniques such as amino acid analysis and high-performance liquid chromatography analysis of culture media.

## 5. Conclusions

Our study highlighted the multifaceted benefits of CSK in promoting skin health through its antioxidant and melanogenesis regulation activities. The observed activation of the Nrf2/HO-1 pathway via MAPKs underscores the cytoprotective mechanisms of CSK against oxidative stress. Notably, CSK exhibits an antipigmentation effect by suppressing genes related to melanin production in melanocytes. These findings strongly suggest the potential of CSK as a novel therapeutic agent in skincare, offering a dual approach to managing oxidative stress and regulating melanin. In the future, our research will focus on identifying the active components of CSK and investigating its antioxidant, anti-melanogenic, and anti-wrinkle and moisturizing effects. Such findings will contribute to solidifying the role of CSK in skincare formulations.

## Figures and Tables

**Figure 1 microorganisms-12-00188-f001:**
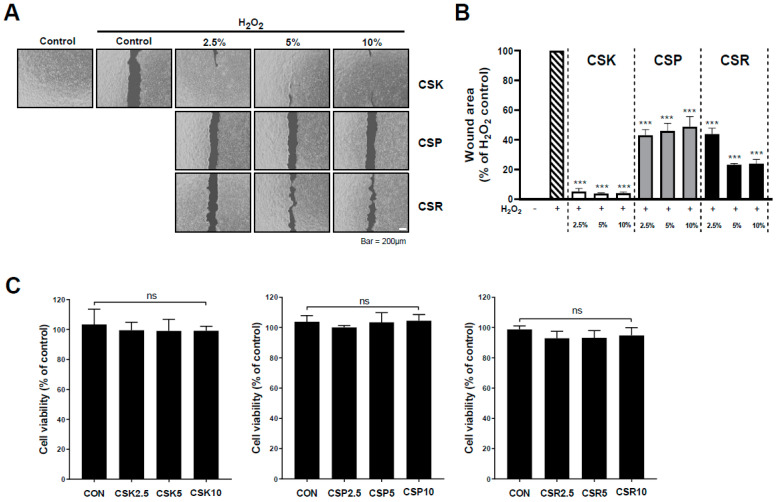
Wound-healing effects of *Lactobacillus* spp. culture supernatant in H_2_O_2_-treated HaCaT cells. (**A**) After a cell-free gap (wound area) was formed, cells were treated with *Lactobacillus* spp. culture supernatant at the indicated concentrations. After 1 h, cells were treated with H_2_O_2_ (500 μM). The wound margin was photographed 24 h after treatment. Scale bar: 200 μm. (**B**) Wound areas were quantified using ImageJ and expressed as the percentage difference between the H_2_O_2_-treated group and the supernatant-treated group. (**C**) HaCaT cells were treated with *Lactobacillus* spp. culture supernatant at the indicated concentrations for 24 h, and cell viability was determined using the MTT assay. Data are presented as the mean ± standard deviation (SD). *** *p* < 0.001, relative to H_2_O_2_-treated cells. ns, not significant; CSP, culture supernatant of *Lactobacillus plantarum*; CSR, culture supernatant of *Lactobacillus reuteri*; CSK, culture supernatant of *Lactobacillus kunkeei*.

**Figure 2 microorganisms-12-00188-f002:**
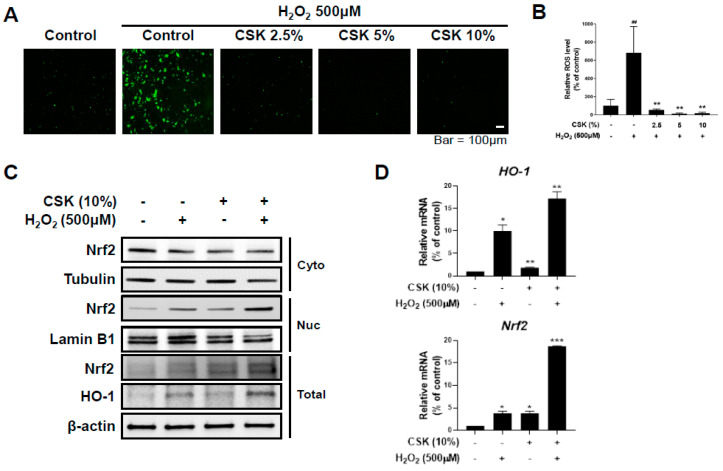
Oxidative stress attenuation by the culture supernatant of *Lactobacillus kunkeei* (CSK) in H_2_O_2_-treated HaCaT cells. (**A**) HaCaT cells were pretreated with CSK for 1 h and then treated with H_2_O_2_ (500 μM) for 3 h. Subsequently, ROS generation was assessed using the dichloro-dihydro-fluorescein diacetate (DCF-DA) assay. Scale bar: 100 μm. (**B**) The fluorescence intensity of DCF was quantified using ImageJ, and data are presented as levels relative to the fluorescence of the control (cells without H_2_O_2_ treatment). (**C**–**E**) HaCaT cells were pretreated with CSK for 1 h and then treated with H_2_O_2_ (500 μM) for 6 h. (**C**) The cytoplasmic, nuclear, and total protein Nrf2 expression and HO-1 total protein expression were examined using Western blotting, and (**D**) mRNA levels were measured using quantitative real-time PCR (qPCR). (**E**) The cells were labeled with anti-Nrf2 and fluorescein isothiocyanate (FITC)-conjugated secondary antibody, and the nuclei were stained with DAPI (4′,6-diamidino-2-phenylindole) and observed under a confocal microscope. Representative images are from at least three independent experiments. Scale bar: 50 μm. Data are presented as the mean ± SD. ## *p*< 0.01, relative to untreated cells. * *p* < 0.05, ** *p* < 0.01, and *** *p* < 0.001, relative to untreated or H_2_O_2_-treated cells.

**Figure 3 microorganisms-12-00188-f003:**
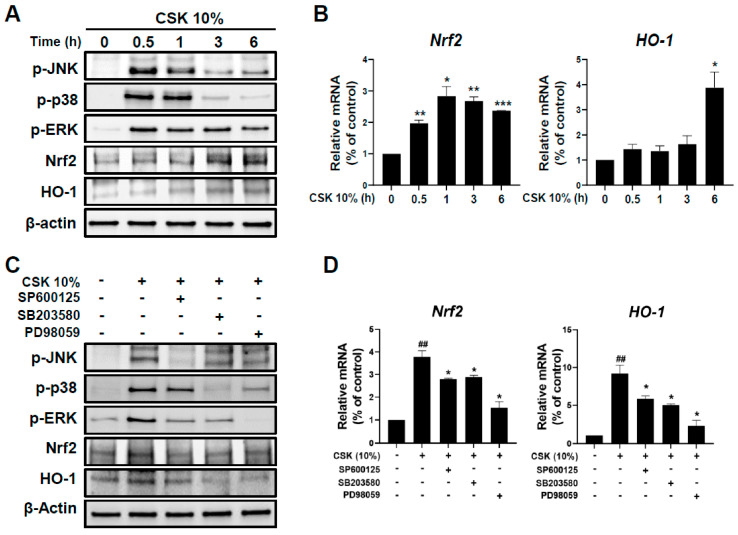
CSK-induced activation of MAPK and NRF2/HO-1 pathways in HaCaT cells. (**A**) HaCaT cells were treated with CSK (10%), and MAPK activation at 0 h, 0.5 h, 1 h, 3 h, and 6 h was examined using Western blotting. Additionally, the Nrf2/HO-1 pathway was examined at 0 h, 0.5 h, 1 h, 3 h, and 6 h. (**B**) HO-1 and Nrf2 mRNA levels at the same time points were measured using qPCR. (**C**) HaCaT cells were pretreated with specific inhibitors, including SP600125 (JNK inhibitor), SB203580 (p38 inhibitor), and PD98059 (ERK inhibitor), for 1 h, followed by treatment with CSK (10%) for 3 h. The activation of the MAPK and Nrf2/HO-1 pathways was examined using Western blotting. (**D**) After CSK (10%) treatment for 12 h, HO-1 and Nrf2 mRNA levels were measured using qPCR. Data are presented as the mean ± SD. ## *p* < 0.01, relative to untreated cells. * *p* < 0.05, ** *p* < 0.01, and *** *p* < 0.001, relative to untreated or CSK-treated cells.

**Figure 4 microorganisms-12-00188-f004:**
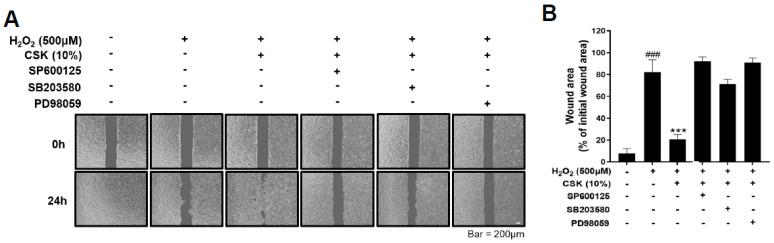
MAPK activation-based wound-healing effect of CSK in H_2_O_2_-treated HaCaT cells. (**A**) The wound area was photographed at 0 h and 24 h after treatment; the cells were first pretreated with inhibitors for 1 h, then treated with CSK (10%) for 1 h, and finally treated with H_2_O_2_ (500 μM) for 24 h. Scale bar: 200 μm. (**B**) Wound areas were quantified using ImageJ and expressed as the percentage difference before and after H_2_O_2_ treatment. Data are presented as mean ± SD. ### *p* < 0.001, relative to untreated cells. *** *p* < 0.001, relative to untreated or H_2_O_2_-treated cells.

**Figure 5 microorganisms-12-00188-f005:**
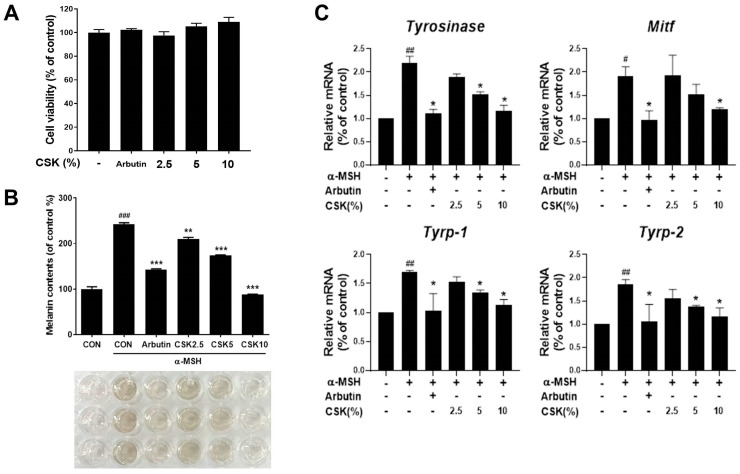
The effects of CSK on melanogenesis in α-melanocyte stimulating hormone (α-MSH)-stimulated B16F10 cells. (**A**) B16F10 cells were treated with arbutin and CSK at the indicated concentrations for 24 h, and cell viability was determined using the MTT assay. (**B**) Antimelanogenic activity was evaluated by quantifying the amount of melanin synthesized in the B16F10 cells. After CSK treatment for 1 h, α-MSH (100 nM) was added to stimulate melanin synthesis. After 72 h, melanin content was measured at 472 nm, using a spectrophotometer. Arbutin (2%) was used as a positive control. (**C**) Melanogenesis-related gene expression was measured after 1 h of CSK treatment, followed by 48 h of α-MSH (100 nM) treatment, using real-time PCR. Data are presented as the mean ± SD. # *p* < 0.05, ## *p* < 0.01, and ### *p* < 0.001, relative to untreated cells. * *p* < 0.05, ** *p* < 0.01, and *** *p* < 0.001, relative to α-MSH-treated cells.

## Data Availability

All data generated or analyzed during this study are included in this published article or in the data repositories listed among the references.

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
