# Peer review of "Antioxidant and Antimelanogenic Activities of Lactobacillus kunkeei NCHBL-003 Isolated from Honeybees"

_microorganisms, 2024, doi:10.3390/microorganisms12010188_

Round 1

Reviewer 1 Report

Comments and Suggestions for Authors

This study assessed the antioxidant and antimelanogenic activities of L. kunkeei culture. The research is noteworthy and current, particularly in its exploration of the dual approach involving the antioxidant activity of L. kunkeei bacteria. The study is well-organized, coherent, and effectively addresses key points. With only minor adjustments needed, I recommend it for publication.

  1. In the introduction, enhance the content by including information about other bacterial species known for their involvement in wound healing, such as L. reuteri and L. plantarum.

  2. In the Materials and Methods section, please clarify whether the probiotic used is a combination of the three bacteria (L. kunkeei, L. reuteri, L. plantarum) or if only L. kunkeei was isolated. If a combination was used, review the results accordingly.

  3. Correct the notation of statistical significance at line 245 and line 304. Specifically, ensure that **p < 0.01 and ***p < 0.001 are accurately represented and distinguishable.

Author Response

Reviewer Comments:
This study assessed the antioxidant and antimelanogenic activities of L. kunkeei culture. The research is noteworthy and current, particularly in its exploration of the dual approach involving the antioxidant activity of L. kunkeei bacteria. The study is well-organized, coherent, and effectively addresses key points. With only minor adjustments needed, I recommend it for publication.

Reviewer 1
1. In the introduction, enhance the content by including information about other bacterial species known for their involvement in wound healing, such as L. reuteri and L. plantarum.

Response) Thank you for your valuable feedback. In response to your recommendation, we have revised the introduction to incorporate additional information. We believe that these enhancements contribute to the overall depth and relevance of the introduction. 

→ In the introduction…

Lactic acid bacteria (LAB), gram-positive bacteria that convert sugar to lactic acid, generate numerous beneficial components [26,27] with anti-inflammatory and antibacterial properties, suppressing the proliferation of harmful bacteria [28]. Researchers are increasingly interested in the treatment of skin wounds using LAB as a treatment with fewer side effects on the skin. Recent studies have shown that Lactobacillus rhamnosus GG and Lactobacillus gasseri, which have been proven to increase gastric barrier repair, accelerate wound healing in primary human keratinocytes [29]. Lactobacillus plantarum supernatant (CSP) has been shown to provide 60-100% protection against H2O2-induced stress in A549 cells; it has also been shown to promote wound healing more quickly than a positive control group in mice [30]. For centuries, bee products, such as honey, propolis, royal jelly, and bee pollen, have been employed to enhance human health and address skin issues [31-35] owing to their anti-inflammatory, anticancer, and antioxidant properties.

  1. In the Materials and Methods section, please clarify whether the probiotic used is a combination of the three bacteria (L. kunkeei, L. reuteri, L. plantarum) or if only L. kunkeei was isolated. If a combination was used, review the results accordingly.

Response) Thank you for your kind comment. We believe that your suggestions and comments are helpful to improve our study. As you suggested, we would like to clarify that the probiotic strains L. kunkeei NCHBL-003, L. plantarum NCHBL-004, and L. reuteri NCHBL-005 were not used in combination. Each strain was individually isolated from the digestive tract of honeybees and used separately in the experiments. The Materials and Methods section has been revised accordingly to reflect this clarification.

→ In the materials and methods…

Preparation of the Culture Supernatants of Lactobacillus spp.

Probiotic strains L. kunkeei NCHBL-003, L. plantarum NCHBL-004, and L. reuteri NCHBL-005 were individually isolated from the digestive tracts of honeybees. In our experiments, each strain was used independently, without any combination treatments.

  1. Correct the notation of statistical significance at line 245 and line 304. Specifically, ensure that **p < 0.01 and ***p < 0.001 are accurately represented and distinguishable.

Response) Thank you for your thoughtful comment. We changed it in figure legends as you pointed out.

→ In the Figure legends…

Figure 2. *p < 0.05, **p < 0.01, and ***p < 0.001, compared to untreated or H2O2-treated cells.

Figure 4. ###p < 0.001, compared with untreated cells. ***p < 0.001, compared to untreated or H2O2-treated cells.

Reviewer 2 Report

Comments and Suggestions for Authors

Comments on the Quality of English Language

English language must be improved

Author Response

Reviewer Comments:
The paper titled "Antioxidant and Antimelanogenic Activities of Lactobacillus kunkeei NCHBL-003 Isolated from Honeybee" investigates the potential therapeutic applications of Lactobacillus kunkeei culture supernatant (CSK) in skincare, focusing on its antioxidative and antimelanogenic properties. The paper provides a significant contribution to the understanding of probiotic-derived bioactive substances in dermatological applications. However, some suggestions could improve it:

Reviewer 2
1. In the abstract section, the authors must highlight the main objective and novelty of the study, which currently appears unclear.

Response) Thank you for your thoughtful review and valuable feedback on the abstract. As suggested by the reviewer, we have made the following revisions.

→ In the abstract…

Excessive reactive oxygen species production can detrimentally impact skin cell physiology, resulting in cell growth arrest, melanogenesis, and aging. Recent clinical studies have found that lactic acid bacteria have a special effect directly or indirectly on skin organs, but the exact mechanism has not been elucidated. In this study, we investigated the mechanisms underlying the antioxidant protective effect and the inhibitory effect on melanin synthesis of Lactobacillus kunkeei culture supernatant (CSK), isolated from Apis mellifera Linnaeus (the western honeybee). CSK exhibited notable efficacy in promoting cell migration and wound healing under oxidative stress, surpassing the performance of other strains. CSK pretreatment significantly upregulated the expression of Nrf2/HO-1 (nuclear factor erythroid 2-related factor 2/heme oxygenase-1), a key player in cellular defenses against oxidative stress, relative to the control H2O2-treated cells. The DCF-DA (dichloro-dihydro-fluorescein diacetate) assay results confirmed that CSK’s ability to enhance Nrf2 and HO-1 expression aligns with its robust ability to remove H2O2-induced reactive oxygen species. Furthermore, CSK upregulated MAPK (mitogen-activated protein kinase) phosphorylation, an upstream signal for HO-1 expression, and MAPK inhibitors compromised the wound-healing effect of CSK. Additionally, CSK exhibited inhibitory effects on melanin synthesis, downregulating melanogenesis-related genes in B16F10 cells. Thus, the present study demonstrated that CSK exhibited antioxidant effects by activating the Nrf2/HO-1 pathway through MAPK phosphorylation, thereby restoring cell migration and demonstrating inhibitory effects on melanin production. These findings emphasize the antioxidant and antimelanogenic potential of CSK, suggesting its potential use as a therapeutic agent promoting wound healing and as an active ingredient in skin-lightening cosmetics.

  1. In the Quantitative Real PCR section, Tables 1 and 2 should be included in the supplementary materials.

Response) Thank you for your kind comment. In response to your suggestion, Table 1 and 2 will be deleted from the manuscript and included as supplementary tables 1 and 2.

→ In the Supplementary Materials …

Table S1: Primer (human) sequences for real time PCR, Table S2: Primer (mouse) sequences for real time PCR.

  1. In the discussion section, the main objective of the study and the literature studies just published on this subject are not clearly articulated. Furthermore, the authors should separate the paragraph about the conclusions and add a paragraph dedicated to future perspectives.

Response) Thank you for your kind comment. We believe that your suggestions and comments are helpful to improve our study. As you suggested, we clearly presented the main objective and deleted or added recent literature studies related to the topic to the discussion. We also separated a paragraph on conclusions and added future perspectives of our findings. . Additionally, recognizing the importance of linguistic accuracy, we have entrusted the English proofreading of the entire manuscript, including references, to a professional service, ESSAYREVIEW (https://essayreview.co.kr/). We believe that these steps will ensure compliance with your guidelines and enhance the overall quality of our submission.

→ In the Discussion…

The numerous health benefits of probiotics are well-recognized, with recent studies particularly emphasizing their positive effects on skin health, such as antioxidant, antipigmentation, and anti-wrinkle properties [45-47]. However, the specific mechanisms underlying these beneficial effects on the skin remain poorly understood. The primary objective of our study was to explore the antioxidant and antimelanogenic properties of L. kunkeei cultures isolated from the honeybee gut. Our focus was on elucidating the mechanism of action of CSK, particularly its impact on the Nrf2/HO-1 activation pathway through MAPK, aiming to understand its role in wound healing via the inhibition of oxidative stress. Additionally, we examined CSK’s effect on α-MSH–induced melanin synthesis inhibition in melanocytes.

…….

Postbiotics have gained considerable interest for their demonstrated efficacy, comparable to probiotic bacteria, making them increasingly relevant in academic research [51]. Recent studies have explored how to enhance skin repair using not only live probiotics but also postbiotics, including heat-killed bacteria and their metabolites. It was found that the fermented supernatant of L. kefiri can postpone cellular senescence by reducing oxidative stress and the expression of aging-related genes in human skin cells affected by H2O2 [52]. Furthermore, the culture supernatant of L. gasseri BNR17 has been shown to inhibit oxidative stress and hyperpigmentation in skin cells [53].

→ In the Conclusion…

Conclusion

Our study highlighted the multifaceted benefits of CSK in promoting skin health through its antioxidant and melanogenesis regulation activities. The observed activation of the Nrf2/HO-1 pathway via MAPKs underscores the cytoprotective mechanisms of CSK against oxidative stress. Notably, CSK exhibits an antipigmentation effect by suppressing genes related to melanin production in melanocytes. These findings strongly suggest the potential of CSK as a novel therapeutic agent in skincare, offering a dual approach to managing oxidative stress and regulating melanin. In the future, our research will focus on identifying the active components of CSK and investigating its antioxidant, anti-melanogenic, and anti-wrinkle and moisturizing effects. Such findings will contribute to solidifying the role of CSK in skincare formulations.

Round 2

Reviewer 2 Report

Comments and Suggestions for Authors

The authors have partially responded to the comments reported

Comments on the Quality of English Language

The English language must be improved